# Optical Bistability in a Tunable Gourd-Shaped Silicon Ring Resonator

**DOI:** 10.3390/nano12142447

**Published:** 2022-07-17

**Authors:** Yishu Chen, Jijun Feng, Jian Chen, Haipeng Liu, Shuo Yuan, Song Guo, Qinghua Yu, Heping Zeng

**Affiliations:** 1Engineering Research Center of Optical Instrument and System, Ministry of Education and Shanghai Key Laboratory of Modern Optical System, School of Optical-Electrical and Computer Engineering, University of Shanghai for Science and Technology, 516 Jungong Rd, Shanghai 200093, China; chenyishu199651@163.com (Y.C.); chenjian1044@163.com (J.C.); liuhaipeng232@163.com (H.L.); ysh183882148@163.com (S.Y.); guosong1745@sina.com (S.G.); 2Key Laboratory of Intelligent Infrared Perception, Shanghai Institute of Technical Physics, Chinese Academy of Sciences, Shanghai 200083, China; 3Chongqing Key Laboratory of Precision Optics, Chongqing Institute of East China Normal University, Chongqing 401120, China; hpzeng@phy.ecnu.edu.cn; 4State Key Laboratory of Precision Spectroscopy, East China Normal University, Shanghai 200241, China

**Keywords:** gourd-shaped resonator, optical bistability, integrated photonics, silicon on insulator, nonlinear phenomenon

## Abstract

In this study, a tunable gourd-shaped ring resonator is demonstrated to generate optical bistability. The system consists of two sub-rings for a gourd shape configuration with a U-shaped wave guiding pathway. The transfer matrix method and FDTD simulation are used to acquire the spectral characteristics of the system. For the fabricated device, the spectra profile and extinction ratio can be effectively tuned by the microheater above the U-shaped waveguide, which matches with the theoretical results. Due to the gourd structure of the resonator, the light waves in two rings can be cross-coupled with each other, and the optical bistability could come out effectively with the change in the input optical power around 6 mW. The presented optical bistability devices have great application potential in optical information processing such as optical storage, switch and logic operation.

## 1. Introduction

Silicon photonic integrated devices have been widely applied for information processing, optical communication and computing due to their complementary metal–oxide semiconductor (CMOS) compatible processing [1,2,3]. Their high refractive index contrast enables a compact footprint, a strong optical field confinement and thus an intense light-matter interconnection, which can lead to efficient generation of nonlinear optical phenomena [4]. Among different nonlinear optics applications, optical bistability has attracted widespread concern in ultrafast signal processing and modulation [5,6]. In addition, a small-scale and steady tuning platform for generating the bistability can be offered by a microcavity, which would be further applied to all-optical modulation, memory, memristor, switch, and so forth [7,8,9,10].

When the light passes through an optical microcavity, a complex nonlinear interaction system may be formed due to self-phase modulation [11], two-photon absorption [12], free-carried absorption [13] or thermo-optic effect [14]. For the silicon ring resonators, optical bistability caused by the two-photon absorption effect has been studied, which could be used for logic gates and suitable for flip-flops’ structure [15,16]. Adjusting the phase of the cavity could also be used to realize optical bistability [17]. To acquire a steady multistability state, many cavities are often essential but with only small output light intensity [18], such as by cascading five identical silicon rings [19]. In order to observe the bistability with low input light intensity, the linear loss of the microring should be as low as possible and the radius of the microring should be as small as possible [15]. Actually, optical bistability can be easily generated through a U-shaped resonator due to the repeated coupling of light waves [20]. In addition, a cross-coupled ring resonator system has been presented to improve the generation efficiency [21,22,23]. However, such a cross-coupled configuration would usually cause a decrease in the resonator’s quality factor, which is not so favorable for the nonlinear optics applications. Optical resonators with a high quality factor can have the capability of trapping and storing photons for longer periods of time, which are obviously favorable for the nonlinear photonics applications. To maintain a high quality factor while taking full advantage of the multiple resonance configuration for wave coupling more times and resonating longer in microcavities, a tunable gourd-shaped silicon ring resonator is proposed here for the efficient generation of optical bistability.

In the following, a tunable gourd-shaped ring resonator system is first discussed and analyzed. The system is composed of a U-shaped waveguide and two ring cavities for a gourd-shape configuration, where the two cavities can realize a high quality factor and the waveguide can consist of a multiple self-coupling structure. By means of the transfer matrix method and FDTD simulation, the spectral properties of the system can be accurately obtained. The device is manufactured by a commercial CMOS foundry, and the measured spectral performance coincides well with the simulation, while the resonant wavelength and extinction ratio can be tuned effectively. Due to the gourd shape configuration of the resonator, optical bistability can be easily realized.

## 2. Device Design

Structure of the proposed gourd-shaped ring resonator is shown in Figure 1a, with Figure 1b for the detailed microscope graph of the fabricated chip and the corresponding magnified views. The silicon waveguide has a width of 450 nm and thickness of 220 nm. The system consists of three directional couplers between the bottom racetrack resonator and the U-shaped wave pathway and that with the top ring, with all having a coupling gap of 250 nm. The corresponding calculated coupling coefficients are 0.6 and 0.9 for a transverse-electrical (TE) polarized wave, respectively. The top ring has a radius of 50 μm, while the bottom racetrack resonator has the same radius and with an additional straight wire 25.6 μm long. The interval between the input and output waveguides is thus set to 127 μm, enabling the chip to couple with the standard fiber array.

The signal light enters from an input port and is divided into two sections after passing through coupler *X* as in Figure 1a. One section of the input light passes through the U-shaped path and arrives at coupler *Y*. The other section of the input light circulates in a counterclockwise (CCW) direction in the racetrack resonator, coupling with the U-shaped path and the top ring resonator at the couplers *Y* and *Z*, respectively, while the light circulates in a clockwise (CW) direction in the top ring and couples back again. The coupled light from coupler *Z* would then interact with the input light at coupler *X*. Different coupling situations would generate different resonance performances and the varying transmittance spectra can be obtained.

Optical transmission properties of the tunable gourd-shaped ring resonator system are first obtained by utilizing the transfer matrix method [24]. Both the CCW and CW mode can be stimulated. $E_{i}^{+\left(-\right)}$ indicates the propagation direction of the electric field in the system, with the plus sign (+) for the CW propagation and the minus sign (−) for the CCW propagation, while that in the directional coupler area between the racetrack resonator and the gourd-shaped cross-coupled ring are denoted by $E_{x_{m}}^{+\left(-\right)}$ and $E_{y_{n}}^{+\left(-\right)}$ (m, n = 1…4), as shown in Figure 1a. The directional coupling area of the tunable gourd-shaped ring resonator is represented by $E_{z_{p}}^{+\left(-\right)}$ (p = 1…4). $c_{r}$ and $t_{r}\mathrm{tr}$ are the crossing-coupling and transmission coefficients, respectively ($c_{r1}^{2}\;$*+*$\;t_{r1}^{2}\;$*=* 1 and $c_{r2}^{2}\;$*+*$\;t_{r2}^{2}\;$*=* 1 for lossless coupling). The transmission T of the system can be deduced by
(1)$$\left[\begin{array}{c} E_{x_{2}}^{+}\\ E_{x_{4}}^{+} \end{array}\right]=\left[\begin{array}{cc} t_{r1} & -ic_{r1}\\ -ic_{r1} & t_{r1} \end{array}\right]\left[\begin{array}{c} E_{x_{1}}^{+}\\ E_{x_{3}}^{+} \end{array}\right],$$
(2)$$\left[\begin{array}{c} E_{y_{2}}^{+}\\ E_{y_{4}}^{+} \end{array}\right]=\left[\begin{array}{cc} e^{-i\theta_{2}} & 0\\ 0 & e^{-i\theta_{1}} \end{array}\right]\left[\begin{array}{c} E_{x_{4}}^{+}\\ E_{x_{2}}^{+} \end{array}\right],$$
(3)$$\left[\begin{array}{c} E_{y_{1}}^{+}\\ E_{y_{3}}^{+} \end{array}\right]=\left[\begin{array}{cc} t_{r1} & -ic_{r1}\\ -ic_{r1} & t_{r1} \end{array}\right]\left[\begin{array}{c} E_{y_{2}}^{+}\\ E_{y_{4}}^{+} \end{array}\right],$$
(4)$$\left[\begin{array}{c} E_{z_{1}}^{+}\\ E_{z_{2}}^{+} \end{array}\right]=\left[\begin{array}{cc} t_{r2} & -ic_{r2}\\ -ic_{r2} & t_{r2} \end{array}\right]\left[\begin{array}{c} E_{z_{3}}^{-}\\ E_{z_{4}}^{+} \end{array}\right],$$
(5)$$\left[\begin{array}{c} E_{z_{4}}^{+}\\ E_{x_{3}}^{+} \end{array}\right]=\left[\begin{array}{cc} e^{-i\theta_{3}} & 0\\ 0 & e^{-i\theta_{3}} \end{array}\right]\left[\begin{array}{c} E_{y_{1}}^{+}\\ E_{z_{2}}^{+} \end{array}\right],$$
and
(6)$$E_{z_{3}}^{-}=e^{-i\theta_{4}}E_{z_{1}}^{+}$$

The electric field of the input light is denoted as $E_{x_{1}}^{+},\;$ while that of the output section is $E_{y_{3}}^{+}$. Transmittance T of system can be calculated by
(7)$$T={\left|t\right|}^{2}={\left|\frac{E_{y_{3}}^{+}}{E_{x_{1}}^{+}}\right|}^{2},$$
with
(8)$$t=-c_{r1}{}^{2}e^{-i\theta_{2}}+t_{r1}{}^{2}e^{-i\theta_{1}}+\frac{A}{B}.$$Here
(9)$$\begin{array}{cc} A= & -c_{r1}^{2}t_{r1}^{2}t_{r2}e^{-i\left(2\theta_{2}+2\theta_{3}\right)}-c_{r1}^{2}t_{r1}^{2}t_{r2}e^{-i\left(\theta_{1}+\theta_{2}+2\theta_{3}\right)}\\ & +c_{r1}^{2}t_{r1}^{2}e^{-i\left(2\theta_{2}+2\theta_{3}+\theta_{4}\right)}+c_{r1}^{2}t_{r1}^{2}e^{-i\left(\theta_{1}+\theta_{2}+2\theta_{3}+\theta_{4}\right)}\\ & -c_{r1}^{2}t_{r1}^{2}t_{r2}e^{-i\left(\theta_{1}+\theta_{2}+2\theta_{3}\right)}-c_{r1}^{2}t_{r1}^{2}t_{r2}e^{-i\left(2\theta_{1}+2\theta_{3}\right)}\\ & +c_{r1}^{2}t_{r1}^{2}e^{-i\left(\theta_{1}+\theta_{2}+2\theta_{3}+\theta_{4}\right)}+c_{r1}^{2}t_{r1}^{2}e^{-i\left(2\theta_{1}+2\theta_{3}+\theta_{4}\right)}, \end{array}$$
and
(10)$$B=1-t_{r2}e^{-i\theta_{4}}-t_{r1}^{2}t_{r2}e^{-i\left(\theta_{2}+2\theta_{3}\right)}+c_{r1}^{2}t_{r2}e^{-i\left(\theta_{1}+2\theta_{3}\right)}\;\;\;+t_{r1}^{2}e^{-i\left(\theta_{2}+2\theta_{3}+\theta_{4}\right)}-c_{r1}^{2}e^{-i\left(\theta_{1}+2\theta_{3}+\theta_{4}\right)}.$$

Here, $\theta_{1}$ is the accumulated phase variation for light wave spreading through the waveguide with a length of $L_{1}$; $\theta_{2}$ and $\theta_{3}$ are these aroused by half the racetrack resonator and one quarter of the racetrack resonator, respectively; $\theta_{4}$ is that aroused by the top ring. The corresponding phase variation can be calculated by
(11)$$\theta_{1}=\frac{2\pi }{\lambda }\left(n_{w}-i\beta_{w}\right)L_{1},$$
(12)$$\theta_{2}=\frac{2\pi }{\lambda }\left(n_{\frac{1}{4}r}-i\beta_{\frac{1}{4}r}\right)L_{2},$$
(13)$$\theta_{3}=\frac{2\pi }{\lambda }\left(n_{\frac{1}{2}r}-i\beta_{\frac{1}{2}r}\right)L_{3},$$
and
(14)$$\theta_{4}=\frac{2\pi }{\lambda }\left(n_{c}-i\beta_{c}\right)L_{4},$$
where λ is the wavelength, $n_{w}$ and $\beta_{w}$ are the real and imaginary sections of the effective mode index of the U-shaped pathway, $n_{\frac{1}{4}r\left(\frac{1}{2}r\right)}$ and $\beta_{\frac{1}{4}r\left(\frac{1}{2}r\right)}$ are those of one quarter of the racetrack resonator and half of the racetrack resonator, respectively, and $n_{c}$ and $\beta_{c}$ are those of the top ring. The imaginary section represents the corresponding propagation loss. $L_{1}$ is the physical waveguide length from area $x_{2}$ to $y_{4}$. $L_{2}$ and $L_{3}$ are the waveguide length of half of the racetrack resonator and one quarter of the racetrack resonator, respectively. $L_{4}$ is the circumference of the top ring.

Changing the cross-coupled coefficients can then obtain different transmission spectra, as shown in Figure 2a for the wavelength range of 1545–1555 nm. The extinction ratio and quality factor at a 1548.56 nm wavelength are 15.05 dB and $2.25\times 10^{5}$, respectively, which performs better than the cross-coupled case [21]. In addition, the performance of the device can be more effectively tuned when an additional phase (from 0 to 0.44π) is applied to the U-shape pathway. Figure 2b presents the spectra corresponding to the additional phase tuning, which can cause periodic resonance wavelength and extinction ratio change. Every 0.02π phase variation can shift the wavelength by about 0.01 nm.

The system performance was further verified by FDTD simulation [25]. Device performance can then be verified by FDTD (Lumerical FDTD Solutions of 8.9.1584). Perfectly matched layer (PML) was adopted for the boundary conditions, while the simulation area was about 165 × 140 μm^2^ with a step size of 0.2 nm. Figure 3 presents the simulation results, while the modeling is upside down for the convenience of light input and output. As shown in Figure 3a, the calculated electric field distribution shows nearly no resonance coupling at a wavelength of 1550 nm, and most of the incident light propagates in the U-shaped pathway. When the wavelength is 1548.56 nm, critical optical coupling happens and the wave can propagate in the gourd-shaped resonator system, as shown in Figure 3b. The obtained results coincide well with the above simulation.

## 3. Device Performance Characterization

As a proof of concept, the designed system was fabricated by a commercial CMOS foundry, by means of a series of thin film deposition [26], photolithography and reactive ion etching (RIE) [27]. For the potential low-cost mass production, traditional e-beam lithography was not adopted here. Figure 4 shows the experimental facility for the device performance characterization. The chip was fiber coupling packaged and placed on a thermal electric cooler (TEC). An amplified spontaneous emission (ASE) light source was used, and the polarization state was adjusted to TE mode by the polarization controller. The output wave was linked to an optical powermeter or optical spectrum analyzer (OSA) with a resolution of 0.01 nm. The chip was monitored by an infrared imaging system. An additional probe table was used for applying the phase-tuning current.

Figure 5 illustrates the measured static spectrum of the fabricated chip, with the inset for an enlarged version at a waveguide of 1550 nm. The coupling loss between the fiber and chip is about 10 dB and the measured spectrum is roughly consistent with the calculated spectrum as shown in Figure 2a. The obtained free spectral range (FSR) is about 1.52 nm, with an extinction ratio of about 6.82 dB, and quality factor of about 1.04 $\times \;10^{5}$ near the 1548.56 nm wavelength. By applying current to the heater above the U-shaped pathway, the effective refractive index can be changed due to the thermal–optical effect, and the spectrum profile can be adjusted effectively, such as with the resonance wavelength and extinction ratio. The spectral response around the 1548.56 nm wavelength with varying the applied voltage is shown in Figure 6a. It can be seen that the resonance wavelength red-shifts while the extinction ratio first increases slightly and then decreases with the applied voltage. When the applied voltage is 1.8 V, the extinction ratio decreased to 1.78 dB with the corresponding resonance wavelength of 1548.62 nm. The tuning performance resembles the simulation results as in Figure 2b. Figure 6b presents the corresponding extinction ratio and resonance wavelength variation with the applied voltage.

It can be seen that the wavelength shift varies in a quadratic relation with the voltage with a voltage of 4.8 applied corresponding to a 0.25 nm shift. This is mainly due to that the effective refractive index varies quasi-linearly with the heat, which has a quadratic connection with the applied voltage. In addition, the extinction ratio changes approximately sinusoidally with the voltage. The device could also be used as an optical switch, e.g., at a working wavelength of 1548.56 nm. If there is no voltage applied, the light is switched off, which would be turned on with a voltage of 1.8, as can be seen from Figure 6a. Considering that the microheater has a resistance of about 970 Ω, the switching power is only 3.34 mW. Though the extinction ratio still needs to be improved, the power consumption is superior to many reported results of about tens of milliwatts [28]. This indicates the potential applications for some inverter optical switches or all-optical logic gates [29,30].

## 4. Optical Bistability Generation

To effectively generate the optical bistability state on the fabricated chip, we used a pump at a wavelength of 1548.86 nm, slightly different from the resonance wavelength for the obvious observation of the influence of input light intensity on the resonance performance [31]. For the experimental setup, the ASE light source in Figure 4 was replaced with a tunable laser and erbium doped fiber amplifier. The output optical power was measured with the change in the input power. Detailed measured results are displayed in Figure 7, with the exclusion of the coupling loss. The output optical power of the system first forms a quasi-linear relation with the input power. Then, the output power falls obviously with the input optical power up to 6.06 mW, and continues a quasi-linear decrease until 7.1 mW. With the increase in input optical power to 11.06 mW, the output power rises correspondingly. As the input optical power decreases to 6.02 mW, the output power declines to 0.082 mW. Then, a sharp rise happens with the continuous fall of the input power. The output power begins to drop again with the input power down to 5.04 mW. Obvious optical bistability could be observed with the change in input optical power.

Optical bistability is more effectively generated by the presented gourd-shaped resonator configuration. Nonlinear optical effects make the waveguide effective refractive index increase with the input light intensity, which leads to a red-shift of the resonance wavelength. It further influences the coupling condition and causes a decline of the light intensity in the ring, which could generate a feedback process due to the multiple resonance for wave coupling more times and resonating longer in microcavities. When the input power is increased to a certain value, the output optical power has a jump, and an optical bistability can be formed. Analogous bistability behavior appears again with the decrease in input light intensity. That is to say, the bistability mainly depends on the light energy stored in the resonator. With the improvement of the quality factor, a gourd-shaped silicon ring resonator can thus generate optical bistability more efficiently. Moreover, the system stability is also checked 20 times by repeated experiments, which showed almost the same performance. The sufficient variation in the input optical power brings about the switching of the output optical power. This behavior displays an important application prospect. Though there is still much space for the improvement of device performance, the presented optical bistability devices can offer the possibility to realize data storage and signal processing with high processing speed and increased bandwidth [31,32,33]. It should also be mentioned that the nonlinear mechanism may include two-photon absorption, Kerr nonlinear effect, free-carrier absorption and dispersion and thermo-optics effects related refractive index change [34], which are relatively complex and needs further in-depth analysis.

## 5. Conclusions

To sum up, a tunable gourd-shaped ring resonator system for an optical bistability generation was designed, fabricated and characterized. The measured transmission spectra are in good agreement with theoretical results, which can be effectively tuned by applying voltage to the heater above the U-shaped pathway. The spectral profile and extinction ratio change periodically with the applied voltage, and an optical switching operation can be realized with a power consumption of about 3.34 mW. Meanwhile, multiple resonance configuration makes the light waves couple more times and resonate longer in microcavities. Thus, the optical bistability can be generated more effectively, which is observed with the variation in input optical power around 6 mW. The presented optical bistability device was fabricated by a commercial CMOS foundry, which can facilitate its applications in all-optical modulation and computing.

## Figures and Tables

**Figure 1 nanomaterials-12-02447-f001:**
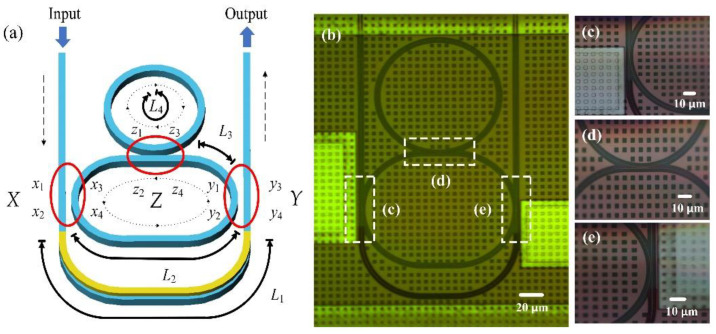
(**a**) Structure view of the tunable gourd-shaped ring resonator system, with dotted line for the light propagating direction. (**b**) Microscopic graph of the chip; (**c**–**e**) are the corresponding magnified views of dotted frame region.

**Figure 2 nanomaterials-12-02447-f002:**
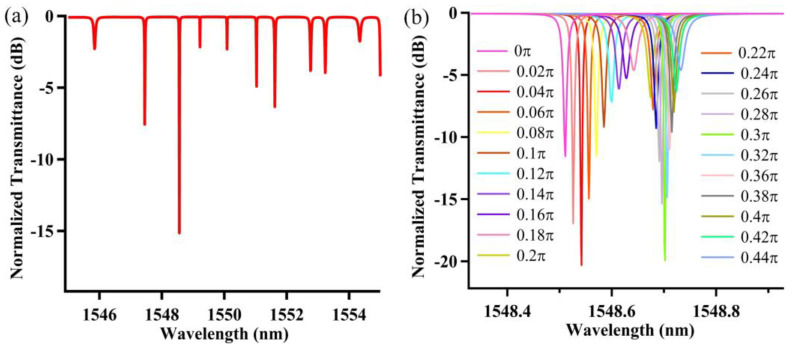
(**a**) Calculated transmission spectrum for the gourd-shaped resonator system, and (**b**) spectrum variation aroused by the additional phase change in the U-shaped pathway with coupling coefficients $c_{r1}\;$= 0.9 and $c_{r2}\;$ = 0.6.

**Figure 3 nanomaterials-12-02447-f003:**
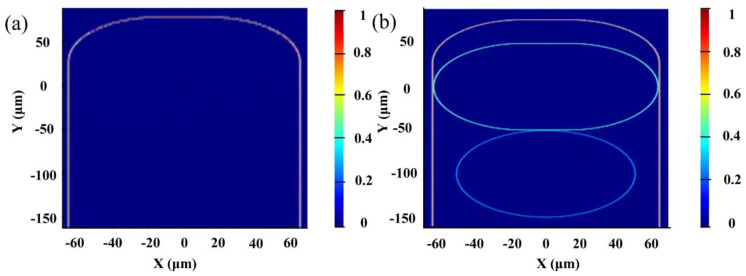
Calculated electric field distribution of the gourd-shaped resonator for (**a**) nearly no resonance coupling at a wavelength of 1550 nm, and (**b**) critical optical coupling at a wavelength of 1548.56 nm.

**Figure 4 nanomaterials-12-02447-f004:**
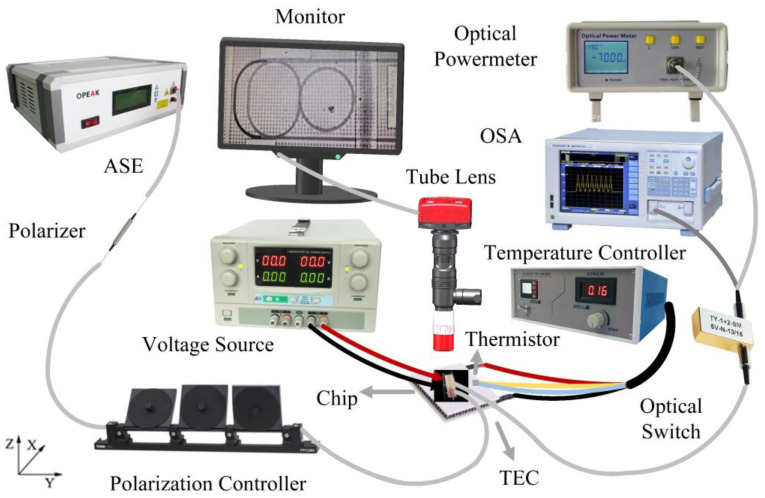
Schematic view of the experimental facility for device performance measurement.

**Figure 5 nanomaterials-12-02447-f005:**
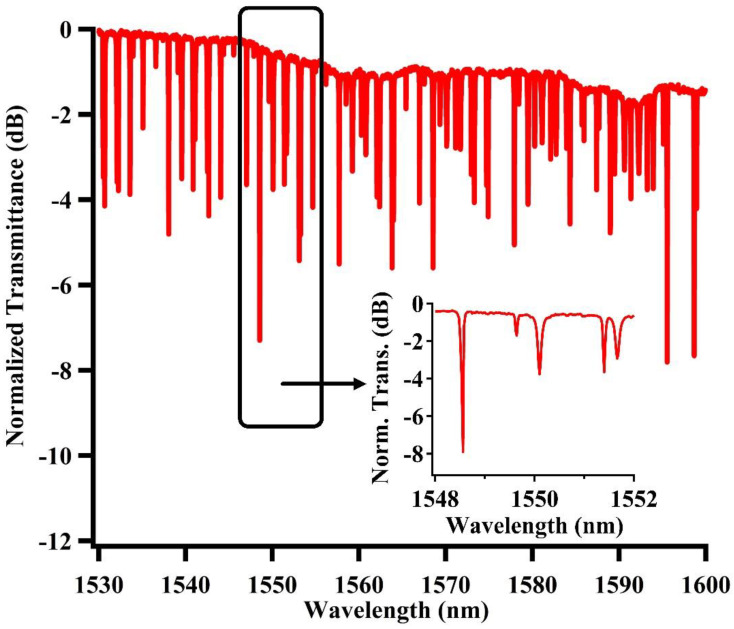
Measured normalized transmission spectrum of the gourd-shaped ring resonator system. Inset: enlarged spectrum around 1550 nm wavelength.

**Figure 6 nanomaterials-12-02447-f006:**
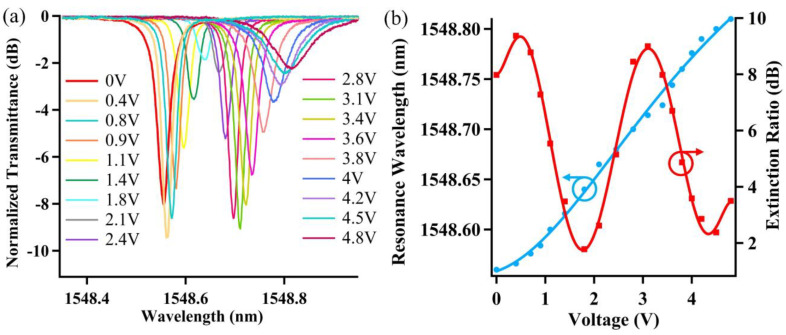
(**a**) Response spectra with changing voltage imposed on the microheater above U-shaped path. (**b**) The corresponding resonance wavelength and extinction ratio variation with the imposed voltage.

**Figure 7 nanomaterials-12-02447-f007:**
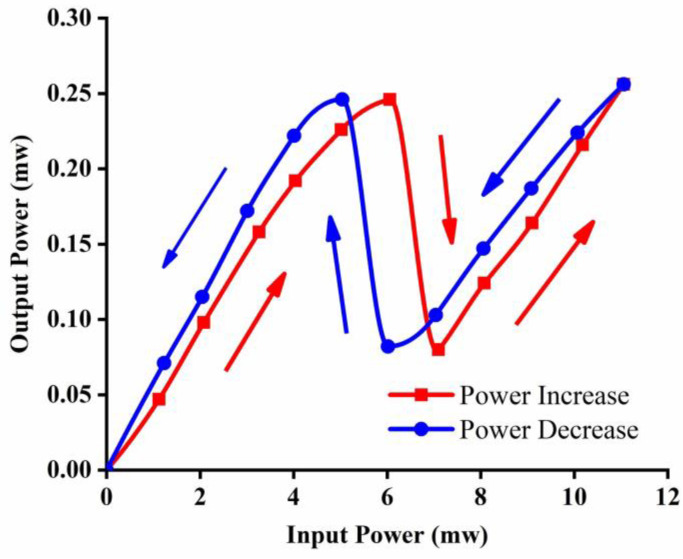
Optical bistability performance of the tunable gourd-shaped ring resonator system.

## Data Availability

The datasets used and/or analyzed during the current study are available from the corresponding author on the reasonable request.

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
