# Peer review of "Optical Bistability in a Tunable Gourd-Shaped Silicon Ring Resonator"

_nanomaterials, 2022, doi:10.3390/nano12142447_

Round 1

Reviewer 1 Report

This paper is interesting but it could be published after minor corrections, mainly, devoted to add additional materials for the comparison of the current results with the state of art. For example, what is the advantage of the new design with the traditional design with the single racetrack resonator?

Some moments that are needing to be commented.

What is the physical reason of choosing the pump wavelength of 1548.86 nm?

Please, provide information about FDTD simulations. The simulation area is very large and it looks like that 2D FDTD method is used. Which software package is used? What is the simulation grid?

The optional will is to look at the comparison of the nonlinear modeling with related experimental verification. May be we will see it in the next paper.

Reviewer 2 Report

The paper under consideration presents the experimental demonstration of the Gourd-shaped Silicon Ring Resonator. However, I have some doubt on the novelty of this work. Can author state the novelty or state-of-the-art of this work that why it is different than previously reported works. And why it deserves a publication?

1) Line 37, "can be" repeated twice.

2) Can author state the novelty of this work. As it seems like a standard CMOS fabrication of a silicon ring resonator. 

3) Why author didnt present any SEM image of the fabricated device?

Round 2

Reviewer 2 Report

I am willing to accept the paper in its current form.